# Using instant messaging applications for consultations in the emergency department: A cross-sectional survey

Sarah S. Abdul-Nabi[1], Souraya Arabi[1], Hind Anan[2], Hussein Hijazi[1], Hani Tamim[3,4], Jean-Marie Al Semaani[2], Maha Makki[3], Afif Mufarrij[1]*

1 Department of Emergency Medicine, American University of Beirut, Beirut, Lebanon, 2 Faculty of Medicine, American University of Beirut, Beirut, Lebanon, 3 Clinical Research Institute, American University of Beirut, Beirut, Lebanon, 4 College of Medicine, Alfaisal University, Riyadh, Saudi Arabia

* am66@aub.edu.lb

## Abstract

### Study objective

This study investigates how consultants and consultees use Instant Messaging Applications (IMAs) during Emergency Department (ED) consultations, examining their attitudes toward these tools and assessing perceptions of existing consultation methods and the perceived need for system improvements.

### Methods

A cross-sectional study was conducted among clinicians involved in Emergency Department consultations at the American University of Beirut Medical Center, Lebanon. Participants completed an online questionnaire assessing their demographics, consultation patterns, IMAs usage, and opinions on consultation modalities. The recruitment period was from 13/11/2023–08/11/2024.

### Results

A total of 120 participants were included (65 consultants, 55 consultees). Consultants were significantly older than consultees (28.7 ± 3.4 vs. 26.6 ± 1.91 years, p < 0.001). While all consultees were residents, 64.6% of consultants were residents and 35.4% were fellows (p < 0.001). All participants reported using smartphones when on call (100%). IMAs were the most preferred consultation method overall (48.3%), followed by smartphone calls (32.5%). Consultants favored smartphone calls significantly more than consultees (49.2% vs. 12.7%, p < 0.001), whereas consultees preferred workstation phone calls (30.9% vs. 0%, p < 0.001). WhatsApp was the dominant IMA used (98.9%). During night shifts, consultants demonstrated a significantly greater reliance on smartphone calls than consultees (84.6% vs. 63.6%, p = 0.008). IMAs were among the most frequently used methods during both day and night calls (80%

**Data availability statement:** The data underlying this study contain potentially identifiable clinical information and cannot be shared publicly due to institutional and ethical restrictions imposed by the American University of Beirut Medical Center (AUBMC) and its Institutional Review Board. Data access may be requested through the AUB Human Research Protection Program (HRPP) / Institutional Review Board at irb@aub.edu.lb. Requests will be evaluated in accordance with AUB's data protection and ethical oversight policies. The dataset used in this study is stored securely within the Department of Emergency Medicine at AUBMC.

**Funding:** The author(s) received no specific funding for this work.

**Competing interests:** The authors have declared that no competing interests exist.

vs. 65.2%), and two-thirds of participants (63.7%) reported concerns regarding legal implications and the need for improved consultation tools.

## Conclusion

The findings highlight the need for structured and secure digital solutions to optimize communication between healthcare teams. With IMAs increasingly embedded in ED consultations, future work should focus on developing dedicated and potentially AI-supported platforms that enhance efficiency, documentation, and data security in clinical communication.

## Introduction

The integration of mobile devices into various professional fields, including healthcare, has transformed medical practice and education. Currently, the majority of medical workers own smartphones (SP) they use often in their clinical routines [1,2]. In fact, a study by Boruff et al. revealed that 93.6% of medical students and residents accessed medical resources and applications on their mobile devices [2]. While only 83 healthcare applications were documented back in 2011 [3], continuous advancements in technology have magnified this number, highlighting the growing reliance on mobile tools in healthcare practice. SP use by healthcare providers significantly enhances clinical decision-making, patient outcomes, and satisfaction, illustrating its potential to improve healthcare quality and efficiency [1,3–5]. Physicians frequently utilize smartphones for oral (56.4%) and written (38.9%) communication, as they find them more efficient than the traditional pagers [1,9]. Studies have shown that mobile devices facilitate faster and more reliable communication, reducing delays in decision-making, and minimizing the risk of medical errors [6–9]. In particular, WhatsApp has gained widespread use in Emergency Departments (EDs) for text messaging, voice notes, and multimedia sharing [10]. It has considerably reduced the patients' length of stay, with a median difference of 30 minutes compared to the standard telephone communication [11].

At the American University of Beirut Medical Center (AUBMC), mobile devices are widely adopted by healthcare providers. In a study conducted at AUBMC, all ED personnel reported owning a mobile device, with most reporting using medical applications in their clinical practice. The majority agreed that mobile devices improved communication and patient care, although fewer felt these tools enhanced teamwork within the department [12]. Instant Messaging Applications (IMAs) such as WhatsApp were widely adopted, though it remains difficult to clearly differentiate between clinical and personal use [12,13].

In the Emergency Department, consultations are a critical component of patient management and typically involve rapid information exchange between ED-based clinicians and consulting services [14]. Consultation requests are commonly initiated by ED teams and may include verbal communication, written documentation, or informal messaging to convey clinical details, urgency, and preliminary assessments. Given

the time-sensitive and high-acuity nature of ED care, efficient communication during consultations is essential to support timely decision-making and coordination between teams. However, variability in communication modalities and practices may influence the clarity, efficiency, and reliability of information transfer during the consultation process [15].

Given the critical role of communication in the ED, where response rates and time are essential, this study aims to investigate how IMAs are used when consulting with medical professionals, exploring their attitudes towards these applications, and identifying areas for improvement in communication practice.

## Methodology

### Study design and setting

This cross-sectional questionnaire-based study was conducted in the Emergency Department at the American University of Beirut Medical Center (AUBMC). Data were collected from 120 ED-based trainees (consultees-55) and residents and fellows from consulting services (consultants-65) between 13/11/2023 and 08/11/2024. AUBMC is an academic tertiary care hospital, handling approximately 55,000 ED visits each year. The ED is staffed by a combination of physicians trained in Emergency Medicine (EM) and experienced practitioners with substantial EM expertise. The EM Residency Program at AUBMC was started in June 2012 as a four-year training program, with an average of five residents per year. In addition to EM residents, residents from other specialties as well as medical students in their fourth year also rotate through the ED as part of their training.

Consultations play a crucial role in patient management in the ED. The ED conducts approximately 18,300 consultations annually, averaging around 50 per day, with varying levels of acuity, ranging from minor injuries to critical conditions requiring immediate intervention. Previously, AUBMC relied on pagers primarily for alert-based communication, such as notifying on-call providers of consultation requests, but later transitioned to a secure, institution-approved electronic system for consultation order placement. Pagers use remain limited within certain services and workflows. In addition, AUBMC provides free Wi-Fi access for staff.

The consultation process begins with an ED trainee evaluating the patient and presenting the case to the ED attending physician, who oversees the initial assessment and workup. If consultation from another service is required, the ED team places a consultation order in the Electronic Medical Record (EMR) to formally document the request. However, this order does not generate an automatic notification to the consulting team.

Therefore, the ED team directly contacts the on-call resident or fellow from the consulting service to communicate the patient information and the clinical question requiring consultation. The consultant subsequently evaluates the patient and discusses the assessment and management plan with the ED team (Fig 1).

As per AUBMC policy, if there is no response after two contact attempts within 10 minutes, the ED attending escalates the request to the consultant service attending-on-call, who becomes responsible for ensuring a prompt response.

### Selection of participants

The study population includes two groups: Consultees and Consultants

| Consultees (covering ED shifts) | Consultants |
|---|---|
| Residents (Emergency Medicine residents and moonlighters (licensed physicians who are not enrolled in a residency or fellowship program at the time of employment and who cover Emergency Department shifts during a transitional "gap year," typically before starting residency or before entering fellowship.), as well as residents rotating in the ED from various specialties: Pediatrics, Internal Medicine, Family Medicine, Otorhinolaryngology, Anesthesiology, Radiology, Obstetrics and Gynecology, Psychiatry and Medical Students. | • Residents: Specialties including Anesthesiology, Dermatology, Radiology, Neurology, Ophthalmology, Otorhinolaryngology, Psychiatry, Surgery (Neurosurgery, General Surgery, Critical Care, Urology, Orthopedics, Plastics), and Obstetrics and Gynecology)<br>• Clinical fellows consulted by ED residents, moonlighters, or students. Subspecialties such as Endocrinology, Gastroenterology, Nephrology, Pulmonology, Infectious Diseases (adult & pediatric), Hematology/Oncology (adult & pediatric), Intensive Care (adult & pediatric), Cardiology (adult & pediatric), and Pediatric Neurology. |

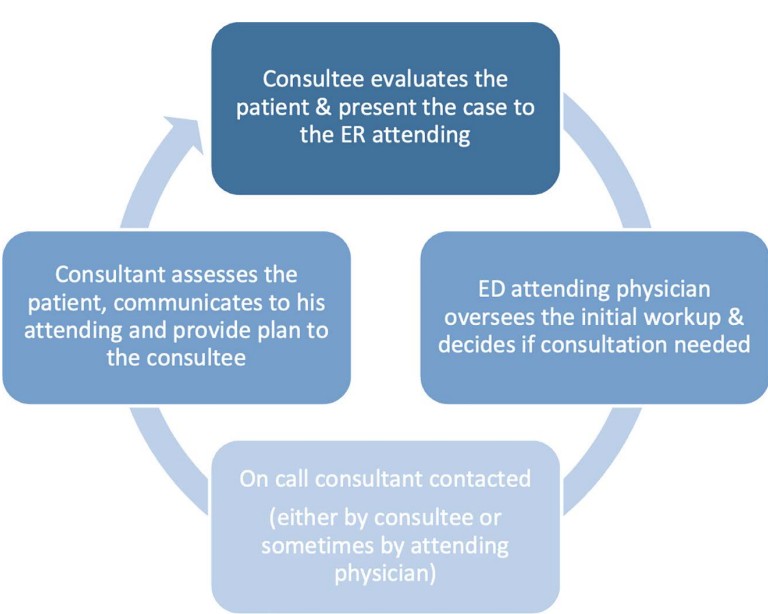

**Fig 1. Consultation process flow chart.**

Inclusion criteria for consultees are all medical students and residents as well as moonlighters covering shifts in the ED. Inclusion criteria for consultants are all residents and clinical fellows consulted by ED residents. There were no exclusion criteria, participation was open to all eligible individuals who consent to take part.

Attending physicians were not included because they supervise, approve, or decline consultations but do not directly engage in the communication process.

In the study, participants are categorized as follow:

• Surgical specialties: General Surgery, Orthopedics, Neurosurgery, OBGYN, Ophthalmology, Otorhinolaryngology

• Acute non-surgical specialties with potential procedures: Radiology, Neurology, Anesthesia, Plumo/ICU, Cardiology, Emergency Medicine

• Acute non-surgical specialties with NO potential procedures: Psychiatry, Pathology, Family Medicine, Nephrology, Gastroenterology, Infectious disease, Endocrinology, Pediatrics, Internal medicine

• Junior Residents: House Physician, Post graduate year (PGY) 1 & PGY2

• Senior Residents: PGY3,4,5,6

## Data collection tool (questionnaire)

A cross-sectional survey was sent by email to residents and fourth-year medical students working in the Emergency Department (consultees), and to residents and fellows from consulting services (consultants) at AUBMC. Attending physicians were not included in the survey as the study focused on trainees who most frequently initiate and receive consultation requests. The recruitment period was from 13/11/2023–08/11/2024. A total of 120 participants completed the questionnaire, including 55 consultees and 65 consultants.

## Data collection procedure

The questionnaire was initially piloted and reviewed by 2–3 consultants and consultees to ensure clarity and relevance before it was distributed. The survey was in English, as all participants demonstrate English proficiency prior to their positions. It explored participant demographics, consultation practices and communication modalities, as well as perceptions related to efficiency, privacy, and areas for improvement in consultation processes. Two tailored survey versions were used: Consultees' Survey (Designed for medical students and residents working in the ED) and Consultants' Survey (Addressed to residents and fellows serving as consultants). Both versions followed similar themes with minor role-specific differences. A copy of the questionnaire developed for this study has been provided as S1 File.

## Ethical consideration

Approval was granted by the Institutional Review Board under protocol number: SBS-2023–0009. This study poses minimal risk to participants, as all survey responses are anonymous and confidential. Participants provided informed online consent before completing the survey. Participation was voluntary, with no identifying information collected.

Additional information regarding the ethical, cultural, and scientific considerations specific to inclusivity in global research is included in the Supporting Information (S2 File).

## Data analysis

Data collection was extracted through SPSS. The Statistical Package for the Social Sciences (SPSS) version 25.0 was used for data cleaning, management, and analysis. Descriptive statistics were presented as mean ± standard deviation (SD) for continuous variables. Categorical data were reported as frequencies and percentages. In the bivariate analysis, the relationships between participant roles (consultant vs consultee) and other categorical variables was analyzed using Pearson's Chi-square **test.** Whereas Student's t-test was used for the association with continuous variables. The statistical significance was set at p-value < 0.05.

## Results

Table 1 presents the Demographics and Professional Characteristics (Residency Program and Level) of Consultants and Consultees. The study included a total of 120 participants, 40% of which were male. These consisted of 65 consultants and 55 consultees. The mean age of the consultants was significantly higher than that of the consultees ($28.7 \pm 3.4$ vs $26.6 \pm 1.9$ years, $p < 0.001$). The distribution of participants across residency programs also showed significant variation ($p < 0.001$). While 25.0% of consultants belonged to surgical specialties, the remaining participants were distributed across acute non-surgical specialties with potential procedures (37.5% consultants vs. 35.8% consultees) and acute non-surgical specialties with no potential procedures (37.5% consultants vs. 37.7% consultees). In terms of residency level, junior residents constituted 41.8% of consultees, which is significantly higher than that of consultants (41.8 vs 23.1%, $p < 0.001$). Conversely, senior residents made up a greater proportion of consultants (56.9%) compared to consultees (32.7%). 20% of consultants were fellows and students accounted for 25.5% of consultee. A significant difference was found in the shift duration when on call where consultants had a longer shift than consultees ($23.1 \pm 7.7$ hours vs. $9.0 \pm 1.9$ hours, $p < 0.001$).

Table 2 presents General Information Related to SP Usage and Consultation Preferences Among Consultants and Consultees. All (100%) participants reported having a SP and using it while on call or duty. Additionally, 16.7% of participants reported that their smartphone was used by another team member for consultation-related communication, more frequently among consultees (23.6%) than consultants (10.8%) (p = 0.06). The preferred method of consultation for consultees was IMAs (48.3%, $p < 0.001$) while consultants preferred phone calls on SP (49.2%). In fact, both groups favored using IMAs like WhatsApp, with 44.6% of consultants and 52.7% of consultees indicating this preference. While 30.9% of consultees preferred using workstation phone calls, none of the consultants selected this method. On the other

**Table 1. Demographics and Professional Characteristics (Residency Program and Level) of Consultants and Consultees.**

| | | Total N = 120 | Consultant N = 65 | Consultees N = 55 | P-value |
|---|---|---|---|---|---|
| Gender | Male | 48 (40.0%) | 29 (44.6%) | 19 (34.5%) | 0.26 |
| **Age** | Mean ± SD | 27.7 ± 3.0 | 28.7 ± 3.4 | 26.6 ± 1.9 | <0.001 |
| **Position** | Resident | 97 (80.8%) | 42 (64.6%) | 55 (100.0%) | <0.001 |
| | Fellow | 23 (19.2%) | 23 (35.4%) | 0 (0.0%) | |
| **Program** | Surgical specialties* | 16 (13.7%) | 16 (25.0%) | 0 (0.0%) | <0.001 |
| | Acute non-surgical specialties with potential procedures** | 43 (37.8%) | 24 (37.5%) | 19 (35.8%) | |
| | Acute non-surgical specialties with NO potential procedures*** | 44 (37.6%) | 24 (37.5%) | 20 (37.7%) | |
| | Students | 14 (12.0%) | 0 (0.0%) | 14 (26.4%) | |
| **Year** | Junior Residents^ | 38 (31.7%) | 15 (23.1%) | 23 (41.8%) | <0.001 |
| | Senior Residents^^ | 55 (45.8%) | 37 (56.9%) | 18 (32.7%) | |
| | Fellows | 13 (10.8%) | 13 (20.0%) | 0 (0.0%) | |
| | Students | 14 (11.7%) | 0 (0.0%) | 14 (25.5%) | |
| **Shift duration when on call (in hours)** | Mean ± SD | 16.6 ± 9.1 | 23.1 ± 7.7 | 9.0 ± 1.9 | <0.001 |

* Surgical specialties: General Surgery, Orthopedics, Neurosurgery, OBGYN, Ophthalmology, Otorhinoloryngology.

** Acute non-surgical specialties with potential procedures: Radiology, Neurology, Anesthesia, Plumo/ICU, Cardiology, Emergency Medicine.

*** Acute non-surgical specialties with NO potential procedures: Psychiatry, Pathology, Family Medicine, Nephrology, Gastroenterology, Infectious disease, Endocrinology, Pediatrics, Internal medicine.

^ Junior Residents: House Physician, PGY1 & PGY2.

^^ Senior Residents: PGY3,4,5,6.

**Table 2. General Information Related to smartphones (SP) Usage and Consultation Preferences Among Consultants and Consultees.**

| | | Total N = 120 | Consultant N = 65 | Consultees N = 55 | P-value |
|---|---|---|---|---|---|
| **Ownership of a SP** | Yes | 120 (100.0%) | 65 (100.0%) | 55 (100.0%) | – |
| **Use of SP when on call/duty** | Yes | 120 (100.0%) | 65 (100.0%) | 55 (100.0%) | – |
| **Smartphone used by another team member** | Yes | 20 (16.7%) | 7 (10.8%) | 13 (23.6%) | 0.06 |
| **Preferred method of consultation** | IMA | 58 (48.3%) | 29 (44.6%) | 29 (52.7%) | <0.001 |
| | Phone call on SP | 39 (32.5%) | 32 (49.2%) | 7 (12.7%) | |
| | Workstation phone call | 17 (14.2%) | 0 (0.0%) | 17 (30.9%) | |
| | Pager | 6 (5.0%) | 4 (6.2%) | 2 (3.6%) | |

hand, consultants preferred using a phone call on their personal smartphones (49.2%), significantly more than consultees (12.7%, p<0.001).

Table 3 presents Consultation Patterns among Consultants and Consultees During Day and Night Calls. For pager use, consultees were significantly more likely to use pagers compared to consultants during both day calls (32.7% vs. 15.6%, p=0.03) and night calls (29.1% vs. 12.3%, p=0.02). Similarly, the use of workstation phone calls differed significantly between the two groups. During day calls, consultees relied more on workstation phone calls compared to consultants (81.8% vs 56.9%, p=0.003). This difference was even more pronounced during night calls, where 80.0% of consultees used workstation phone calls compared to only 49.2% of consultants (p<0.001). Regarding phone calls made via SP, a significant difference was observed during night calls, with a higher proportion of consultants using this method compared

**Table 3. Consultation Patterns among Consultants and Consultees during Day and Night Call.**

| | | Total N=120 | Consultant N=65 | Consultees N=55 | P-value |
|---|---|---|---|---|---|
| **Pager** | Day call | 28 (23.5%) | 10 (15.6%) | 18 (32.7%) | 0.03 |
| | Night call | 24 (20.0%) | 8 (12.3%) | 16 (29.1%) | 0.02 |
| **Workstation phone call** | Day call | 82 (68.3%) | 37 (56.9%) | 45 (81.8%) | 0.003 |
| | Night call | 76 (63.3%) | 32 (49.2%) | 44 (80.0%) | <0.001 |
| **Phone call from SP** | Day call | 93 (77.5%) | 53 (81.5%) | 40 (72.7%) | 0.25 |
| | Night call | 90 (75.0%) | 55 (84.6%) | 35 (63.6%) | 0.008 |
| **IMA** | Day call | 96 (80.0%) | 53 (81.5%) | 43 (78.2%) | 0.65 |
| | Night call | 75 (62.5%) | 38 (58.5%) | 37 (67.3%) | 0.32 |

to consultees (84.6% vs. 63.6%, *p*=0.008). Although the overall use of IMA did not significantly differ between consultants and consultees during day or night calls.

Table 4 presents Usage Patterns and Frequency of IMA for Consultations among Consultants and Consultees. WhatsApp was the most frequently used Instant Messaging Application IMA for consultations, with nearly all participants (98.9%) relying on it, and no significant differences observed between consultants and consultees (p=0.33). Text messaging was the most used consultation tool via IMA, with 95.6% of participants favoring it, and no significant difference between consultants and consultees (97.9% vs 93.2%, p=0.28). Voice messages were also widely used (82.4%) by both consultants and consultees. Moreover, images were used by consultees more than consultants (63.6% vs 42.6%, p=0.04). The average number of consultations conducted via IMA per call or duty was slightly higher among consultees compared to consultants though this difference was not statistically significant (6.45±7.42 vs. 4.94±4.21, p=0.23). Regarding language use in IMA consultations, English was the predominant language for both groups (80.2%), with no significant variation between consultants and consultees (p=0.33).

Table 5 presents participants opinion regarding the use of IMA for consultations. In terms of trust, text messages remained the most trusted tool (70.3%). However, consultants were significantly more likely to trust image-based consultations than consultees (29.8% vs 4.5%, p=0.002). On the other hand, voice messages were trusted by 40.4% of consultants but only 8.6% of consultees, reflecting a significant difference (p=0.001). Voice messages were reported as the most distrusted tool by 44.0% of participants, followed by images (24.2%), with no significant differences between groups. When rating the utility of IMA-based consultations on a range of 0% (Never) to 100% (Always), participants rated IMA as fast (84.4±23.6), easy (87.9±24.0), and accessible (88.5±19.5), with no major differences between groups. However, consultants perceived IMA consultations as significantly more distracting than consultees (58.5±35.1 vs 38.6±33.9, p=0.007).

Finally, 45.3% agree that the current consultation process is adequate, while 73.6% believe it should be improved or better organized, and two-thirds (66.0%) agree that an electronic platform or application should be developed for consultations.

## Discussion

The study included 120 participants, with 65 consultants and 55 consultees. Consultants were older than consultees, and while all consultees were residents, 35.4% of consultants were fellows. The participants' residency programs varied significantly, with consultants more likely to belong to surgical specialties. All participants used SPs while on duty, with Instant Messaging Applications being the most used consultation modality. Consultants favored SP calls, whereas consultees relied more on workstation phone calls. Consultants also had significantly longer shift durations. Consultation patterns varied between day and night shifts, with consultees more likely to use pagers and workstation phone calls, while consultants

Table 4. Usage Patterns and Frequency of IMA for Consultations among Consultants and Consultee.

| | | Total N = 120 | Consultant N = 65 | Consultees N = 55 | P-value |
|---|---|---|---|---|---|
| **Most used IMA** | WhatsApp | 90 (98.9%) | 46 (97.9%) | 44 (100.0%) | 0.33 |
| | Other | 1 (1.1%) | 1 (2.1%) | 0 (0.0%) | |
| **Most USED method** | Text Message | 87 (95.6%) | 46 (97.9%) | 41 (93.2%) | 0.28 |
| | Image | 48 (52.7%) | 20 (42.6%) | 28 (63.6%) | 0.04 |
| | Video | 11 (12.1%) | 6 (12.8%) | 5 (11.4%) | 0.84 |
| | Voice Message | 75 (82.4%) | 42 (89.4%) | 33 (75.0%) | 0.07 |
| **Average number of IMA consultations per call/shift** | Mean ± SD | 5.7 ± 6 | 4.9 ± 4.2 | 6.4 ± 7.4 | 0.23 |
| **Language used via IMA** | English | 73 (80.2%) | 36 (76.6%) | 37 (84.1%) | 0.33 |
| | Arabic | 3 (76.6%) | 3 (6.4%) | 0 (0.0%) | |
| | Chat language (Arabic written in English letters) | 15 (16.5%) | 8 (17%) | 7 (15.9%) | |

preferred SP calls, particularly at night. WhatsApp was the most used IMA, and text messaging was the predominant consultation tool. Consultants reported greater trust in image-based and voice message consultations than consultees. Overall, the study highlights significant differences in consultation preferences and communication patterns between consultants and consultees.

The study focused on directly comparing consultees and consultants within a single ED system, using two role-specific versions of the same questionnaire. The aim was to capture differences in workflow, preferences, and perceived challenges. Unlike most prior work, which focuses primarily on general smartphone use or single-group assessments, this study differentiates the consultation behavior of ED-based trainees from that of consulting services [16]. It also incorporates day- versus night-shift communication patterns, perceived trust across multiple IMA modalities, and the impact of shift duration, all features that are rarely examined simultaneously in previous investigations. This dual-group, workflow-specific approach provides a clearer understanding of how consultation dynamics differ across levels of training and service responsibilities within the ED environment.

In our study, consultants showed a markedly stronger preference for SP calls compared to consultees, who preferred workstation phone calls. This may be due to differences in workflow and physical location, with consultants often off-site and consultees consistently positioned in the ED [18]. This difference was even more pronounced during night shifts, when consultants showed a significantly higher preference for SP calls. A likely explanation is that consultants are often covering multiple units or off-site locations during night duty, making direct smartphone communication more practical and responsive than workstation-based methods. This pattern is consistent with prior studies showing that clinicians rely more heavily on mobile phone calls during after-hours or when managing wider geographic responsibilities [6,19]. WhatsApp was the dominant IMA used by nearly all participants, likely because it is universally accessible, familiar, and reliable for rapid clinical communication [17]. The extensive use of IMAs observed in both groups aligns with earlier studies reporting broad uptake of asynchronous communication tools among clinicians [15]. In addition, the preference for phone calls aligns with studies noting increased smartphone reliance during longer or more demanding shifts [16].

Unlike previous work that shows a uniform adoption of WhatsApp [13], our results demonstrate a more nuanced pattern, with consultants showing greater trust in image-based and voice-message consultations than consultees. This might be due to images offering clearer clinical detail for triage accuracy and voice messages allowing rapid, hands-free communication when consultants are mobile or covering multiple units mechanisms previously described in studies examining remote clinical communication dynamics [20]. The predominance of text messaging in our sample reinforces earlier observations that asynchronous communication remains a cornerstone of clinical coordination [17,18]. Yet consultants in our study reported more distraction, connection issues, and diagnostic limitations than consultees. Mobile communication

**Table 5. Consultants and Consultees opinion regarding the use of IMA for consultations.**

| | | | Total N = 120 | Consultant N = 65 | Consultees N = 55 | P-value |
|---|---|---|---|---|---|---|
| **Most TRUSTED method** | Text Message | | 64 (70.3%) | 35 (74.5%) | 29 (65.9%) | 0.37 |
| | Image | | 16 (17.6%) | 14 (29.8%) | 2 (4.5%) | 0.002 |
| | Video | | 3 (3.3%) | 3 (6.4%) | 0 (0.0%) | 0.24 |
| | Voice Message | | 22 (26.8%) | 19 (40.4%) | 3 (8.6%) | 0.001 |
| **Most DISTRUSTED method** | Text Message | | 16 (17.6%) | 8 (17.0%) | 8 (18.2%) | 0.88 |
| | Image | | 22 (24.2%) | 15 (31.9%) | 7 (15.9%) | 0.07 |
| | Video | | 12 (13.2%) | 8 (17.0%) | 4 (9.1%) | 0.26 |
| | Voice Message | | 40 (44.0%) | 22 (46.8%) | 18 (40.9%) | 0.57 |
| **Consultation Request through IMA causes any legal problems** | Yes | | 58 (63.7%) | 31 (66.0%) | 27 (61.4%) | 0.25 |
| **For the below questions: Answers range from 0% (Never) to 100% (Always)** | | | | | | |
| **Consultation request through IMA is:** | **Fast** | Percentage | 84.4 ± 23.6 | 84.2 ± 26.0 | 84.6 ± 21.0 | 0.93 |
| | **Easy** | Percentage | 87.9 ± 24.0 | 87.8 ± 24.9 | 88.1 ± 23.2 | 0.95 |
| | **Reliable** | Percentage | 69.8 ± 26.2 | 67.5 ± 28.5 | 72.2 ± 23.6 | 0.40 |
| | **Understandable** | Percentage | 75.8 ± 24.8 | 73.4 ± 25.8 | 78.4 ± 23.9 | 0.34 |
| | **Accessible** | Percentage | 88.5 ± 19.5 | 87.8 ± 20.1 | 89.2 ± 19.0 | 0.73 |
| | **Professional** | Percentage | 51.1 ± 28.1 | 50.0 ± 31.3 | 52.3 ± 24.6 | 0.70 |
| | **Convenient** | Percentage | 80.8 ± 24.7 | 80.8 ± 22.8 | 80.7 ± 26.9 | 0.97 |
| | **Distracting** | Percentage | 48.9 ± 35.7 | 58.5 ± 35.1 | 38.6 ± 33.9 | 0.007 |
| **Connection issue delaying consultation via IMA** | Percentage | | 23.6 ± 19.8 | 28.2 ± 22.5 | 18.8 ± 15.4 | 0.02 |
| **The current Consultation Request Tool is adequate** | Disagree | | 19 (17.9%) | 10 (18.2%) | 9 (17.6%) | 0.26 |
| | Neutral | | 39 (36.8%) | 24 (43.6%) | 15 (29.4%) | |
| | Agree | | 48 (45.3%) | 21 (38.2%) | 27 (52.9%) | |
| **The current Consultation Request Tool should be improved/more organized** | Disagree | | 2 (1.9%) | 1 (1.8%) | 1 (2.0%) | 0.79 |
| | Neutral | | 26 (24.5%) | 15 (27.3%) | 11 (21.6%) | |
| | Agree | | 78 (73.6%) | 39 (70.9%) | 39 (76.5%) | |
| **It is necessary to develop an electronic platform or an application exclusively for consultation request** | Disagree | | 18 (17.0%) | 12 (21.8%) | 6 (11.8%) | 0.35 |
| | Neutral | | 18 (17.0%) | 8 (14.5%) | 10 (19.6%) | |
| | Agree | | 70 (66.0%) | 35 (63.6%) | 35 (68.6) | |

has been shown to increase interruptions and workflow disruption among clinicians, especially when they are managing multiple responsibilities across different clinical areas [21]. In addition, studies on mobile messaging systems report that unstable connections and inconsistent Wi-Fi coverage can undermine communication reliability, particularly for clinicians who move frequently between hospital units [22]. Diagnostic limitations of remote communication have been documented, with studies showing that text- and image-based exchanges may omit key contextual cues, reducing diagnostic certainty and increasing the risk of misinterpretation in complex assessments [23].

While IMAs enhance speed and accessibility, they raise concerns regarding confidentiality, documentation, and medico-legal exposure. This echoes prior findings emphasizing inconsistent record-keeping and the absence of standardized guidelines [24]. Studies in other clinical settings have shown that implementing structured communication tools improves the clarity of consult requests and reduces perceived miscommunication [25], and that teaching structured methods that enhance the content and clarity of interprofessional referrals [26]. The results highlight a pressing need for streamlined, secure communication systems capable of integrating clinical data, supporting documentation, and reducing variability across teams. Future research should evaluate how structured digital platforms, potentially incorporating

advanced automation or AI-supported triage, could improve workflow reliability, enhance data protection, and standardize communication across consultees and consultants within the ED.

## Limitations

This study has several limitations. First, the cross-sectional design limits the ability to establish causal relationships or assess changes over time. The study is also restricted to a single institution, AUBMC, with a small sample size of only 120 participants, which may limit the generalizability of the findings to other settings or larger, more diverse populations. Furthermore, attending physicians were not included in the participant population, which may affect the comprehensiveness of the findings regarding communication dynamics. In fact, some ED attendings physicians who may opt to contact the attending consultant directly without going through the channel of residents. Moreover, some consultations are conducted in person when a consultant is already present and evaluating another patient. Rather than utilizing formal communication methods such as IMA or phone calls, ED physicians often choose face-to-face interactions with consultants to inform them of additional consults. While this direct interaction facilitates immediate discussion, faster decision-making, and more efficient patient management, it may reduce the potential for delays associated with electronic or phone-based communication. Finally, the study did not explore whether participants had used other mobile applications prior to using the WhatsApp for consultations, which may have influenced their preferences and attitudes toward WhatsApp.

## Supporting information

**S1 File. Study questionnaire.** Full survey instruments used for consultants and consultees assessing consultation practices and use of instant messaging applications in the emergency department.
(DOCX)

**S2 File. Inclusivity in global research.** Completed PLOS checklist detailing ethical, cultural, and authorship considerations for research conducted at AUBMC.
(DOCX)

## Author contributions

**Conceptualization:** Hani Tamim, Afif Mufarrij.

**Data curation:** Sarah S Abdul-Nabi, Souraya Arabi, Hussein Hijazi, Maha Makki, Afif Mufarrij.

**Formal analysis:** Hani Tamim, Maha Makki.

**Methodology:** Hussein Hijazi, Afif Mufarrij.

**Project administration:** Afif Mufarrij.

**Supervision:** Afif Mufarrij.

**Validation:** Afif Mufarrij.

**Writing – original draft:** Sarah S Abdul-Nabi, Souraya Arabi, Hind Anan, Hussein Hijazi, Jean-Marie Al Semaani.

**Writing – review & editing:** Sarah S Abdul-Nabi, Souraya Arabi, Hussein Hijazi, Jean-Marie Al Semaani.

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
