## [Decision Letter · Decision Letter 0]

12 Dec 2025

Dear Dr. Mufarrij,

Thank you for submitting your manuscript to PLOS ONE. After careful consideration, we feel that it has merit but does not fully meet PLOS ONE’s publication criteria as it currently stands. Therefore, we invite you to submit a revised version of the manuscript that addresses the points raised during the review process.

We look forward to receiving your revised manuscript.

Kind regards,

Li V. Yang, Ph.D.

Academic Editor

PLOS One

Journal Requirements:

4. In the online submission form you indicate that your data is not available for proprietary reasons and have provided a contact point for accessing this data. Please note that your current contact point is a co-author on this manuscript. According to our Data Policy, the contact point must not be an author on the manuscript and must be an institutional contact, ideally not an individual. Please revise your data statement to a non-author institutional point of contact, such as a data access or ethics committee, and send this to us via return email. Please also include contact information for the third party organization, and please include the full citation of where the data can be found.

Reviewers' comments:

Reviewer's Responses to Questions

**Comments to the Author**

1. Is the manuscript technically sound, and do the data support the conclusions?

Reviewer #1: Partly

Reviewer #2: Partly

2. Has the statistical analysis been performed appropriately and rigorously?

Reviewer #1: Yes

Reviewer #2: Yes

3. Have the authors made all data underlying the findings in their manuscript fully available?

Reviewer #1: Yes

Reviewer #2: Yes

4. Is the manuscript presented in an intelligible fashion and written in standard English?

Reviewer #1: Yes

Reviewer #2: No

Reviewer #1: Very important study that explores a crucial aspect of patient care related to communication across medical teams in the Emergency Departments with consulting services.

Abstract:

1. Study objective:

a. the use of consultants is confusing here as you state “residents and consultants”. I suggest using it only to refer the teams being consulted and refer to the team doing the consulting as consultee. Your aim then is about how “consultants and consultees” within the ED use IMA etc

b. One stated objective is to identify areas for improvement. How is the study designed to identify this? If this is not included in the survey questionnaire, then remove from objective

2. Methods: Need to mention that this is within an ED context.

3. Result: some key results related to perceptions are missing from here.

4. Conclusion: this needs to be supported by the study findings. Use of AI as main part of the conclusion is not supported by findings.

Introduction:

1. In general it makes a good argument for increasing usage of mobile devices in health care. Since the study focuses on communication around consultation practices, the background needs a section on that specifically. What are current practices around initiating consultation request and sharing information around the consult between the consulting teams and consultants? This needs to be added, ideally in the ED context if available, given the focus of the study.

2. Authors state that whatsap has “gained widespread use in the ED” , this needs a source/reference

3. In the paragraph about AUBMC, authors state that “all ED personnel owned”. Grammar does not fit here. If this is based on a study, then state “In a study conducted at AUBMC, all personnel reported owning …”

4. Last paragraph related to aims, I suggest remove whatsapp to remain consistent across the article (with abstract).

Methods section

1. The study design needs to be better outlined.

• This is a cross sectional survey, what type of survey?

• Dates conducted

• Include who was surveyed

2. Study setting:

a. The paragraph relating to how you place consults is important but needs some work:

i. What type of communication were pagers used for?

ii. What is Webex system? I suggest use of type of communication rather than brand and some description of what that system entails (security, access etc)

iii. The statement about free wifi and use of smartphone being more “convenient” sounds like it is an assumption. Suggest to just state that the institution provides free wifi.

iv. The paragraph related to consultation process:

1. states that “if a specialist consultation is required”. EM physicians are also specialists. I suggest usage of “if a consultation from another service is required”.

2. The last paragraph is a good description of policy and process and clearer than the paragraph before it where it is not clear whether the ED team contacts the attending consultant or the trainee teams on the consultant service. Figure should reflect that process outlined in last paragraph.

3. Do not use company name (EPIC), state electronic medical record system and briefly describe features related to consultation (is there an inter-team communication module within the EMR).

b. Selection of Participants:

i. First sentence states that an email was sent “ to all residents, fourth year medical students…and consultants.” Does “consultants” here mean attending physicians or consultant services? Please remain consistent in use of word consultants across the document. A few sentences down you state that it was sent to consultees in the ED and consultants. First sentence should be consistent with latter in terms of who was surveyed.

ii. Was the survey sent to all the trainees in the ED and all trainees across consultant services? What about attending physicians were they included? From table it seems they were not included. If they weren’t then this should be clarified, and WHO speficially was surveyed needs to be consistent across the entire document (from abstract all the way across the paper).

iii. You mention “moonlighters” – what are these? Attendings, residents. Please clarify

iv. What is total number surveyed in each group (consultants and consultee)

v. The domains explored “ demographics, consultation process, method of communication and perspectives on privacy” do not necessarily align with objectives stated. Suggest aligning with each other more clearly.

Results:

Pager usage is reported but in setting section authors report that pagers were phased out. Please explain in background, where in the organization pagers continue to be part of communication system.

Authors report that Webex is the formal process for consultation but the tables do not clearly reflect where this category is.

Table 2. Shift duration when on call may be better placed in Table 1 since it is not about consultation preferences.

Table 5. Most used method – shouldn't this go in the utilization table (table 4) since it is not an “opinion”.

Discussion

First paragraph is a good summary

Second and third, fourth paragraph. It is not clear what the study findings are and what is pulled from literature. Please restate your study finding related to the focus of the paragraph before the comparative literature review to clarify what you added to the literature and how it compares to prior findings.

Conclusion related AI is not supported by the study. Can be mentioned as a modality to be explored but not listed as main recommendation drawn from study findings.

Reviewer #2: Dear Author,

After a careful review, I would like to offer the following comments and suggestions to help improve the clarity, organization, and overall quality of the paper:

1. The methodology section should be reorganized to follow a standard and coherent structure.

2. The “Study Population” subsection should include only the eligibility criteria of participants; study tools or instruments should not be described in this section.

3. A separate and detailed description of the data collection tool should be included.

4. The data collection tool should be referred to as a “questionnaire” rather than a “survey” (as indicated in the manuscript).

5. The data collection process needs to be described more clearly, outlining the specific procedures followed. Additionally, statistical analysis should be presented in a distinct “Data Analysis” subsection.

6. In-text citations should be formatted according to the journal’s specified referencing style.

7. In discussion section, present your study’s findings explaining how your results align with or differ from existing evidence, but do so without describing other studies in detail.

.

Reviewer #1: No

Reviewer #2: **Yes:** Dr. Tanima AhmedDr. Tanima AhmedDr. Tanima AhmedDr. Tanima Ahmed

---

## [Author Response · Author response to Decision Letter 1]

4 Feb 2026

RESPONSE TO REVIEWERS

Reviewer #1

Abstract:

1. Study objective:

a. the use of consultants is confusing here as you state “residents and consultants”. I suggest using it only to refer the teams being consulted and refer to the team doing the consulting as consultee. Your aim then is about how “consultants and consultees” within the ED use IMA etc

Response: Thank you for this comment. We have revised the abstract to consistently use functional roles, referring to ED-based clinicians initiating consultations as “consultees” and clinicians from other services as “consultants,” and removed role-based terms (e.g., residents) from the study objective for clarity and consistency across the manuscript.

b. One stated objective is to identify areas for improvement. How is the study designed to identify this? If this is not included in the survey questionnaire, then remove from objective

Response: Thank you for this comment. We agree that the original wording could imply a broader exploratory aim. We have revised the study objective to explicitly reflect the questionnaire-based assessment of participants’ perceptions of current consultation tools and their perceived need for improvement, ensuring alignment between the stated objectives and the study design.

2. Methods: Need to mention that this is within an ED context.

Response: Thank you for this comment. We have revised the Methods section of the abstract as well as the manuscript to explicitly state that the study was conducted within the Emergency Department consultation context.

3. Result: some key results related to perceptions are missing from here.

Response: Thank you for this suggestion. We have expanded the Results section of the abstract to include key perception-related findings, including participants’ views on efficiency, legal concerns, and the perceived need to improve current consultation tools.

4. Conclusion: this needs to be supported by the study findings. Use of AI as main part of the conclusion is not supported by findings.

Response: Thank you for this comment. We have revised the Conclusion to ensure that it is directly supported by the study findings.

⸻

Introduction:

1. In general it makes a good argument for increasing usage of mobile devices in health care. Since the study focuses on communication around consultation practices, the background needs a section on that specifically. What are current practices around initiating consultation request and sharing information around the consult between the consulting teams and consultants? This needs to be added, ideally in the ED context if available, given the focus of the study.

Response: Thank you for this comment. We have revised the Introduction to include a focused paragraph describing current consultation communication practices within the Emergency Department context, highlighting how consultation requests are initiated and how clinical information is typically shared between teams. This addition strengthens the background and aligns it more closely with the study’s focus.

2. Authors state that whatsap has “gained widespread use in the ED” , this needs a source/reference

Response: Thank you for this comment. We have added an appropriate reference to support the statement regarding the widespread use of WhatsApp in Emergency Department communication.

3. In the paragraph about AUBMC, authors state that “all ED personnel owned”. Grammar does not fit here. If this is based on a study, then state “In a study conducted at AUBMC, all personnel reported owning …”

Response: Thank you for this comment. We have revised the sentence to correct the grammar and to explicitly clarify that this statement is based on a previously published study conducted at AUBMC.

4. Last paragraph related to aims, I suggest remove whatsapp to remain consistent across the article (with abstract).

Response: Thank you for this comment. We have removed “WhatsApp” from the paragraph.

⸻

Methods section

1. The study design needs to be better outlined.

• This is a cross sectional survey, what type of survey?

• Dates conducted

• Include who was surveyed

Response: Thank you for this comment. We have revised the Study Design section to more clearly outline the study methodology by explicitly stating the questionnaire-based cross-sectional design, specifying the data collection period, and summarizing the participant groups surveyed.

2. Study setting:

a. The paragraph relating to how you place consults is important but needs some work:

i. What type of communication were pagers used for?

ii. What is Webex system? I suggest use of type of communication rather than brand and some description of what that system entails (security, access etc)

iii. The statement about free wifi and use of smartphone being more “convenient” sounds like it is an assumption. Suggest to just state that the institution provides free wifi.

Response: Thank you for your comment. We have revised the Study Setting section to improve clarity and neutrality by specifying the role of pagers as alert-based communication tools, removing the use of brand names in favor of a generic description of an institution-approved electronic communication system, and eliminating interpretive language regarding convenience. The text now states only that the institution provides free Wi-Fi access for staff.

**iv. The paragraph related to consultation process:

1. states that “if a specialist consultation is required”. EM physicians are also specialists. I suggest usage of “if a consultation from another service is required”.**

Response: Thank you for your comment. We have revised the wording to specify that consultations refer to services outside the Emergency Department, aligned the description of the consultation workflow with the institutional policy.

2. The last paragraph is a good description of policy and process and clearer than the paragraph before it where it is not clear whether the ED team contacts the attending consultant or the trainee teams on the consultant service. Figure should reflect that process outlined in last paragraph.

Response: Thank you for this comment. We have ensured that Figure 1 accurately reflects the consultation process as outlined in the policy paragraph.

3. Do not use company name (EPIC), state electronic medical record system and briefly describe features related to consultation (is there an inter-team communication module within the EMR).

Response: Thank you for this comment. We have revised the manuscript to remove the company name and now refer generically to the electronic medical record (EMR) system when describing the formal consultation process.

⸻

b. Selection of Participants:

i. First sentence states that an email was sent “ to all residents, fourth year medical students…and consultants.” Does “consultants” here mean attending physicians or consultant services? Please remain consistent in use of word consultants across the document. A few sentences down you state that it was sent to consultees in the ED and consultants. First sentence should be consistent with latter in terms of who was surveyed.

ii. Was the survey sent to all the trainees in the ED and all trainees across consultant services? What about attending physicians were they included? From table it seems they were not included. If they weren’t then this should be clarified, and WHO speficially was surveyed needs to be consistent across the entire document (from abstract all the way across the paper).

iii. You mention “moonlighters” – what are these? Attendings, residents. Please clarify

iv. What is total number surveyed in each group (consultants and consultee)

v. The domains explored “ demographics, consultation process, method of communication and perspectives on privacy” do not necessarily align with objectives stated. Suggest aligning with each other more clearly.

Response: Thank you for these comments. We have revised the Selection of Participants section to ensure consistent use of terminology by clearly defining consultees as ED-based trainees and consultants as residents and fellows from consulting services. We have explicitly stated that attending physicians were not included in the survey, clarified the role of moonlighters, and specified the total number of participants in each group. In addition, we revised the description of the questionnaire domains to more closely align with the stated study objectives.

⸻

Results:

Pager usage is reported but in setting section authors report that pagers were phased out. Please explain in background, where in the organization pagers continue to be part of communication system.

Authors report that Webex is the formal process for consultation but the tables do not clearly reflect where this category is.

Table 2. Shift duration when on call may be better placed in Table 1 since it is not about consultation preferences.

Table 5. Most used method – shouldn’t this go in the utilization table (table 4) since it is not an “opinion”.

Response: Thank you for these comments. We have clarified that pagers, although largely phased out, remain in limited use within certain services, which explains their continued reporting in the Results. We also clarified that the electronic medical record system is used for formal consultation orders, whereas the tables reflect communication modalities, accounting for the absence of this category in the tables. In addition, we moved “shift duration when on call” to Table 1 and relocated utilization-related variables from Table 5 to Table 4 to improve clarity and logical organization.

⸻

Discussion:

First paragraph is a good summary

Second and third, fourth paragraph. It is not clear what the study findings are and what is pulled from literature. Please restate your study finding related to the focus of the paragraph before the comparative literature review to clarify what you added to the literature and how it compares to prior findings.

Conclusion related AI is not supported by the study. Can be mentioned as a modality to be explored but not listed as main recommendation drawn from study findings.

Response: Thank you for your comments. We have revised the Discussion to clearly distinguish our study findings from prior literature by explicitly restating key results at the beginning of each paragraph before comparative discussion. In addition, we have revised the Conclusion and Discussion to frame AI-based solutions as potential future areas for exploration rather than conclusions drawn from the present findings.

⸻

Reviewer #2

1. The methodology section should be reorganized to follow a standard and coherent structure.

Response: Thank you for this comment. We have reorganized the Methodology section to follow a standard and coherent structure, clearly separating the study design and setting, study population, data collection tool, data collection procedure, and data analysis.

2. The “Study Population” subsection should include only the eligibility criteria of participants; study tools or instruments should not be described in this section.

Response: Thank you for this comment. We have revised the Study Population subsection to include only participant eligibility criteria and role definitions, and removed all descriptions related to the study tool and data collection procedures, which are now presented in separate subsections.

3. A separate and detailed description of the data collection tool should be included.

Response: Thank you for this comment. A separate subsection describing the data collection tool has now been added, providing a clear and detailed description of the questionnaire, including its development, structure, and domains assessed.

4. The data collection tool should be referred to as a “questionnaire” rather than a “survey” (as indicated in the manuscript).

Response: Thank you for this comment. We have revised the manuscript to consistently refer to the data collection tool as a “questionnaire” throughout the Methods section and the remainder of the manuscript.

5. The data collection process needs to be described more clearly, outlining the specific procedures followed. Additionally, statistical analysis should be presented in a distinct “Data Analysis” subsection.

Response: Thank you for this comment. We have clarified the data collection process by presenting it in a dedicated “Data Collection Procedure” subsection that outlines the distribution method, recruitment period, and participant inclusion. In addition, we have presented all statistical methods in a distinct “Data Analysis” subsection to improve clarity and organization.

6. In-text citations should be formatted according to the journal’s specified referencing style.

Response: Thank you for this comment. We have reviewed the manuscript and ensured that all in-text citations are consistently formatted according to the journal’s specified referencing style.

7. In discussion section, present your study’s findings explaining how your results align with or differ from existing evidence, but do so without describing other studies in detail.

Response: Thank you for this comment. We have revised the Discussion section to more clearly foreground our study’s findings and to explain how they align with or differ from existing evidence, while limiting detailed descriptions of prior studies and keeping the literature comparison concise.

---

## [Decision Letter · Decision Letter 1]

13 Mar 2026

Dear Dr. Mufarrij,

Thank you for submitting your manuscript to PLOS ONE. After careful consideration, we feel that it has merit but does not fully meet PLOS ONE’s publication criteria as it currently stands. Therefore, we invite you to submit a revised version of the manuscript that addresses the points raised during the review process.

We look forward to receiving your revised manuscript.

Kind regards,

Li V. Yang, Ph.D.

Academic Editor

PLOS One

Journal Requirements:

Reviewers' comments:

Reviewer's Responses to Questions

**Comments to the Author**

Reviewer #1: All comments have been addressed

2. Is the manuscript technically sound, and do the data support the conclusions?

Reviewer #1: Yes

3. Has the statistical analysis been performed appropriately and rigorously?

Reviewer #1: Yes

4. Have the authors made all data underlying the findings in their manuscript fully available?

Reviewer #1: Yes

5. Is the manuscript presented in an intelligible fashion and written in standard English?

Reviewer #1: Yes

Reviewer #1: Thank you for addressing all the comments.

A few additional comments to improve the article:

In the section on study design:

1. The second sentence is redundant and can be removed. “This is a cross-sectional study conducted at ..”

2. In the methods section that describes the consulting process, the authors mention that and order is placed followed by the ED team contacting the consultant team. What is the goal of this second contact? Is this where patient information is shared and the question to the consultant is outlined? Or is that done through the order? Please clarify..

Table 2:

What does use of one’s smartphone by someone else mean?

.

Reviewer #1: No

---

## [Author Response · Author response to Decision Letter 2]

18 Mar 2026

1. The second sentence is redundant and can be removed. “This is a cross-sectional study conducted at ..”

Thank you for the comment. The sentence has been removed and the paragraph was revised to improve clarity and conciseness.

2. In the methods section that describes the consulting process, the authors mention that and order is placed followed by the ED team contacting the consultant team. What is the goal of this second contact? Is this where patient information is shared and the question to the consultant is outlined? Or is that done through the order? Please clarify..

Thank you for the comment. We revised the Methods section to clarify the consultation workflow. Specifically, we clarified that a consultation order is placed in the EMR for documentation and tracking purposes but does not generate an automatic notification to the consulting team. The ED team subsequently contacts the on-call consultant directly using any method of the ones mentioned in the manuscript to communicate patient information and the clinical question.

Table 2

What does use of one’s smartphone by someone else mean?

Thank you for the comment. We clarified the wording of this variable in Table 2. It refers to situations in which another team member uses the participant’s smartphone to communicate with consulting services during clinical duties.

---

## [Editor Report · Decision Letter 2]

26 Mar 2026

Using Instant Messaging Applications for Consultations in the Emergency Department: A cross-sectional survey

PONE-D-25-46511R2

Dear Dr. Mufarrij,

We’re pleased to inform you that your manuscript has been judged scientifically suitable for publication and will be formally accepted for publication once it meets all outstanding technical requirements.

Kind regards,

Li V. Yang, Ph.D.

Academic Editor

PLOS One
---

## [Editor Report · Acceptance letter]

PONE-D-25-46511R2

PLOS One

Dear Dr. Mufarrij,

I'm pleased to inform you that your manuscript has been deemed suitable for publication in PLOS One. Congratulations! Your manuscript is now being handed over to our production team.

Kind regards,

on behalf of

Dr. Li V. Yang

Academic Editor

PLOS One